# Romantic Love and Reproductive Hormones in Women

**DOI:** 10.3390/ijerph16214224

**Published:** 2019-10-31

**Authors:** Piotr Sorokowski, Agnieszka Żelaźniewicz, Judyta Nowak, Agata Groyecka, Magdalena Kaleta, Weronika Lech, Sylwia Samorek, Katarzyna Stachowska, Klaudia Bocian, Aleksandra Pulcer, Agnieszka Sorokowska, Marta Kowal, Katarzyna Pisanski

**Affiliations:** 1Institute of Psychology, University of Wrocław, 50-529 Wrocław, Poland; agata.groyecka@gmail.com (A.G.); 264699@uwr.edu.pl (M.K.); weronikalech93@gmail.com (W.L.); sylwia.samorek@gmail.com (S.S.); kasia.stachowska@gmail.com (K.S.); klaudiastorky@gmail.com (K.B.); ola.pulcer@gmail.com (A.P.); sorokowska@gmail.com (A.S.); marta7kowal@gmail.com (M.K.); 2Department of Human Biology, University of Wrocław, 50-529 Wrocław, Poland; agnieszka.zelazniewicz@uwr.edu.pl (A.Ż.); judyta.nowak@uwr.edu.pl (J.N.)

**Keywords:** love, sex hormones, fecundity, luteinizing hormone (LH), follicle-stimulating hormone (FSH), testosterone (T)

## Abstract

Increased reproductive success is among the most commonly proposed adaptive functions of romantic love. Here, we tested if hormonal changes associated with falling in love may co-vary with hormonal profiles that predict increased fecundity in women. We compared blood serum levels of estradiol (E2, E2/T), luteinizing hormone (LH), follicle-stimulating hormone (FSH), prolactin (PRL), free testosterone (fT), and cortisol (CT), measured in the early follicular phase of the menstrual cycle in single women (*N* = 69) and in women at the beginning of a romantic heterosexual relationship who reported being in love with their partner (*N* = 47). Participants were healthy, regularly cycling women aged 24 to 33 who did not use hormonal contraception. We found that women in love had higher levels of gonadotropins (FSH, LH) and lower testosterone levels compared to single women who were not in love. These groups of women did not, however, differ in terms of estradiol, prolactin, or cortisol levels.

## 1. Introduction

Falling in love is accompanied by many easily distinguishable physiological changes in the body and brain [1]. These physiological and neural changes can influence behavior and impact various physiological processes in humans [1,2,3,4]. Most studies examining hormonal mechanisms in relation to love have focused on pair bonding and commitment [5,6,7,8,9,10], and have thus largely neglected the role of hormones on other aspects of human relationships, such as the phenomenon of falling in love Yet, the hormones most commonly targeted by such studies (e.g., oxytocin, prolactin, testosterone, and cortisol) are known to be involved in a range of behaviors and socioemotional contexts. For instance, oxytocin (OT), a pleiotropic hormone commonly known to influence bonding and attachment in humans [11,12], affecting not only behaviors related to reproduction, food intake, and inflammatory responses [13,14,15] but also gonadotropin release, thus possibly influencing activity of the hypothalamic–pituitary–gonadal axis (HPG) [16]. Therefore, a rise in OT related to experiencing the ‘sensation’ of love and close physical contact with a beloved partner [17] will likely evoke other physiological effects. Similarly, prolactin (PRL), besides its primary function in lactation, influences social behavior and fertility [2,18,19]. It is therefore plausible that proximity to a beloved partner, evoking strong romantic feelings and thus physiological responses, may also trigger a boost in hormones related to fecundity.

To the authors’ knowledge, only one study to date has focused on changes in fertility or reproductive hormones related to the experience of falling in love ([1]; see also [20] for testosterone). In that study, Marazziti and Canale [1] reported higher cortisol levels among men and women who had fallen in love within the past six months compared to those who had not (single and in long-term relationships). The researchers attributed these group differences in hormone levels to the arousal (and stress) accompanying the formation of a new social bond. Further, men who had recently fallen in love also had lower testosterone levels than controls, whereas testosterone levels were relatively elevated among women in love. Finally, the researchers showed a promising, though non-significant, trend in their small sample of women (*N* = 12), wherein women who had recently fallen in love had non-significantly higher levels of estradiol than control subjects, and thus were potentially more fecund. 

The aim of the present study was to test, in a larger and more representative sample of women (compared to [1]), whether passionate love influences hormonal profiles in a way that can increase female fecundity. 

We tested if hormone levels measured during the early follicular phase of the menstrual cycle in women in a new romantic relationship (up to 6 months), who declared that they were in love, differed from those of single women who had not been in a romantic relationship for at least 6 months and declared not being in love. Hormone samples included: (1) Estradiol (E2) level, a key hormone in predicting female fecundity [21], and its ratio to testosterone level (E2/T); (2) gonadotropin levels (FSH, LH), key components of HPG functioning and women’s reproductive functioning [22]; and (3) prolactin level, which is important for reproductive health and is regulated by dopamine levels [23]. We also measured cortisol (C) and free testosterone (fT) levels in all women due to the possible impact of these hormones on the functioning of the HPG axis [24,25], and because previous studies have elicited contradictory results in terms of reporting either decreased [20] or increased [1] levels of testosterone in women who had recently fallen in love compared to a control group. Here, we also controlled for women’s general and current health, age, and body-mass-index (BMI), all of which can affect fecundity. We predicted that healthy women in a new romantic relationship, whose hormone levels are within a normal range and who have no diagnosed fertility problems, should exhibit hormonal profiles associated with an enhanced probability of conception, including higher levels of gonadotropins, estradiol, and prolactin, and higher estradiol/testosterone ratios (E2/T) in comparison to single women who are not in love. 

## 2. Materials and Methods 

The study was conducted in accordance with the Declaration of Helsinki and all aspects of the study were approved by the Ethical Review Board of the Institute of Psychology at the University of Wroclaw. All participants provided written informed consent and received a small gift as compensation for their involvement.

### 2.1. Participants

Participants were recruited via local press releases, radio, internet, and television advertisements. They were informed that the study would focus on an association between relationship status and levels of hormonal and health markers. Women who showed an interest in taking part at this stage then received detailed descriptions of the study procedures, including questionnaires and blood sampling protocol. 

Study inclusion criteria were as follows: women aged 25 to 30 years, in good health (i.e., lack of current infection or chronic diseases), with no diagnosed or self-reported fertility problems, no use of antibiotics or anti-inflammatory drugs within the past two weeks, not taking hormonal contraception or hormone treatment currently or within the past three months, a BMI (calculated based on self-reported weight and height) within the normal range (<30 kg/m^2^), no children, and self-assessed heterosexuality. General health status was controlled based on self-reported health and using basic clinical markers, such as blood morphology with a smear and high sensitive C-reactive protein levels (hsCRP), a non-specific marker of systemic inflammation. Participants showing evidence of ongoing infection (leucocytes count above the normal range or hsCRP levels >10 mg/L) were excluded from further analyses. All participants had normal-range hormone levels (i.e., levels of hormones that fluctuate during the menstrual cycle were appropriate for the early follicular phase).

A final sample of 116 women who met these criteria volunteered to take part in the study. Sixty-nine of the women reported being single and not currently in love (“single sample”), and forty-seven declared that they were in love (“love sample”). Single women were at least 6 months without a romantic partner and declared that they were not in love with any (even a virtual) person. As we targeted the hormonal changes accompanying the initial phase of love, only women at the beginning of their partnership (minimum 3 months to maximum 6 months in a relationship) were included in the “love sample”. Self-reported “in love” status was further confirmed by evaluating the level of passion, assessed using Sternberg’s Triangular Love Scale (STLS questionnaire, Polish adaptation [26]). We used the passion subscale of the STLS because passion (or ‘eros’) typically plays a central role in falling in love [27,28] and is a particularly critical component of love in the context of human reproductive success [29,30,31].

### 2.2. Procedure

Testing included two phases. In the recruitment phase, conducted online, participants completed several questionnaires, including the Passion Subscale of the STLS [26]. All participants who met the study inclusion criteria (see above) were then invited to the second phase of the study. They were informed that blood measurements would need to be taken between the second and fourth day of their menstrual cycle (early follicular phase), and were scheduled accordingly. Scheduled participants were reminded not to consume any food or caffeine within 12 h prior to arriving for the second phase of the study and were encouraged to avoid stressful situations and extraneous physical exertion on the day of testing. They were also asked to refrain from alcohol consumption, drugs, and heavy meals the day before testing. This second phase of the study involved an individual meeting with an experimenter and blood collection from each individual. Prior to blood sampling, participants completed a questionnaire concerning their current health in order to exclude participants with any symptoms of infection. 

### 2.3. Blood Collection, Preparation, Storage, and Testing 

Fasting blood serum samples were taken during the early follicular phase (between the second and fourth day) of the menstrual cycle from all women. Blood specimens were collected by venipuncture in the morning hours between 7:30 and 9:30 am into commercially available tubes containing anticoagulant (BD Vacutainer®). Basic morphological blood tests (blood morphology with smear) were carried out by a commercial professional analytical laboratory in Wrocław, Poland. Blood samples for hormonal and inflammatory marker analyses were transported to the laboratory at the University of Wrocław within 4 h of collection, wherein serum samples were immediately separated by centrifugation, portioned into sterile microtube (SARSTEDT®), and frozen at –70 °C for further hormonal and inflammatory marker analysis.

From the serum samples, we measured estradiol (E2), luteinizing hormone (LH), follicle-stimulating hormone (FSH), prolactin (PRL), free testosterone (fT), and cortisol (CT) levels. We also computed the E2 to T ratio (E2/T). Serum hormonal and hsCRP concentrations were detected by solid phase enzyme-linked immunosorbent assay using commercial kits from DEMEDITEC ® (DE4399 for estradiol, DE1289 for LH, DE1288 for FSH, DE1291 for prolactin, DE2924 for free testosterone, DE1887 for cortisol, and DE740011). Intra- and inter-assay coefficients of variation and assay sensitivity based on the manufacturer’s instructions were, respectively, used: (1) For estradiol: <7.87%; <8.78%; <1.39 pg/mL; (2) for LH <7.62%; <11.02%; =1.27 mIU/mL; (3) for FSH >7.91%; >7.18%; =0.856 Miu; (4) for prolactin <5.87%; <6.22%; =0.35 ng/mL; (5) for free testosterone <10%; <10%; =0.06 pg; (6) for cortisol <8.1%; <7.7%; =2.5 ng/mL; and (7) for hsCRP: <6.9%; <6.3%; =0.02 µg/mL. Serum samples were assayed in duplicate in accordance with the kit instructions. Absorbance was measured using a spectrophotometer Asys340 (Biochrom®). Hormone or hsCRP concentrations were calculated in relation to the standard curves and expressed in pg/mL for estradiol and free testosterone, mIU/mL for FSH and LH, ng/mL for prolactin and cortisol, and µg/mL for hsCRP.

### 2.4. Statistical Analyses

To determine the required sample size, we used G*Power software (version 3.1) (Heinrich-Heine-University, Düsseldorf, Northrhine-Westphalia, Germany) [32]. For planned comparisons involving t-tests, to obtain a power of 0.80 with an alpha level set to 0.05 and to observe an effect of the average size (*d =* 0.5), the projected sample size was 102 individual subjects.

A normal distribution of hormone levels was achieved by log-transformation. We used two-tailed independent sample t-tests to compare the levels of all analyzed hormones between women who reported being in love, characterized by a high level of passion (*N* = 47), and single women who were not in love (*N* = 69). Further, we conducted a series of linear regressions to examine the relationships between romantic relationship status (single versus in love) and hormone levels, controlling for women’s BMI and age. Hormone levels of all participants fell within the normal range. Analyses were performed in Statistica 12.0 (Statsoft, Kraków, Poland). All effects were deemed statistically significant if *p* < 0.05.

## 3. Results

Compared to single women, women who reported being in love had higher levels of FSH and LH, and lower levels of free testosterone (fT) (see Table 1). We found no differences in estradiol (E2), prolactin (PRL), or cortisol levels (CT) between single women and those in love (though group differences in E2/T levels approached significance). 

Control variables (BMI or age) did not differ significantly between groups and the results of tthet linear regressions were unaffected after controlling for these factors (Table 2). Linear regressions further confirmed significant or marginally significant positive relationships between women’s LH, FSH, T, and E2/T levels and their reported state of being in love.

## 4. Discussion

We compared the blood serum levels of estradiol (E2 and E2/fT), luteinizing hormone (LH), follicle-stimulating hormone (FSH), prolactin (PRL), free testosterone (fT), and cortisol (CT), measured in the early follicular phase of the menstrual cycle in single women who were not in love and women in the beginning of romantic relationships who reported a high degree of passionate love. We found that women in love had higher levels of gonadotropins (FSH, LH) and lower free testosterone (fT) levels compared to single women but did not differ in terms of estradiol (E2), prolactin (PRL), or cortisol (CT). 

We also observed a trend of higher estradiol to testosterone ratios (E2/T, *p* = 0.06) in women in love compared to single women; however, this association was mainly driven by higher testosterone levels in single women (those not in love) compared to women in love. A possible increase in the serum E2/T ratios of women in love is interesting as it could suggest increased aromatization of serum T by peripheral tissues. While this possibility should be considered in future studies, here it is highly speculative and the increase, though robust, failed to achieve statistical significance. 

Increased gonadotropin levels and potentially higher E2/T levels among women in love and at the early stage of their romantic relationship may reflect ongoing increases in HPG axis activation. In healthy women of reproductive age this can result in increased ovarian activity and increased estradiol synthesis [33], wherein higher estradiol levels could translate to higher fecundity [21]. It is possible that this is a developing effect and that a delayed response from the ovaries, compared to HPG activation, may explain the lack of significant differences in estradiol levels between the groups of women, which may manifest at a later stage. However, previous research has shown that the apparent influence of romantic love on the HPA axis can occur very early in the relationship (i.e., in the first two weeks) [4]. Adaptive increases in fecundity during a romantic relationship should likewise occur relatively early, increasing the chances of successful conception in the early stages of a relationship. 

Higher fecundity in women in love might also be reflected in higher progesterone levels in the mid-luteal phase of the menstrual cycle; however, progesterone was not measured in this study. Increases in progesterone levels during the second phase of the cycle are strongly associated with women’s fecundity, allowing for successful conception and implantation [34]. This mid-luteal rise in progesterone concentration is part of an adaptive ovarian response to unfavorable conditions, temporarily modulating reproductive functions [35]. Indeed, progesterone levels in women of reproductive age are highly sensitive and responsive to changes in various environmental, lifestyle, dietary [35], and behavioral- factors, including parental behavior [36] or social closeness [37]. This presumption, however, requires further research, employing longitudinal, repeated measurements of estradiol and gonadotropin levels as well as luteal progesterone levels during the first stages of romantic bond development. Another possibility is that, despite the critical role of the HPG axis in a woman’s fertility, it interacts within a complex hormonal milieu and the impact of romantic love on fertility should be further tested on a wider battery of physiological factors related to fecundity and other more comprehensive albeit crude fertility measures, such as time to pregnancy. Alternatively, the increased level of gonadotropins observed here may be merely related to hypothalamic involvement in the process of falling in love [38], not resulting in higher fecundity.

A lack of differences in prolactin levels between the two groups of women tested here may be related to a shorter time of prolactin reactivity in response to emotional states [39] and a relatively wider range of prolactin in conception cycles, indicating a relatively low impact of prolactin level within its clinical range on a woman’s fecundity [40].

The results of our study further corroborate previous reports showing different levels of testosterone in single women and those in love. Similar to the results of van Anders and Goldey [20], we found lower free testosterone levels in women in love than in single women. This result is in line with observations of lower testosterone associated with long-term relationships and parenting in both men and women [41]. It is possible that passionate love effectively lowers testosterone levels early in a relationship, which may in turn also increase fecundity [42]. On the other hand, Marazziti and Canale [1] showed contradictory results and higher levels of testosterone in women in long-term relationships compared to a control group of single women. The researchers’ inclusion of women in long-lasting relationships (rather than those in the early stages of a relationship) may largely account for these conflicting results. Future investigations may test whether changes in hormones, such as testosterone, may mediate the relationship between love and loyalty and even monogamy and attachment of the female partner in a couple.

Although differences in the testosterone levels of single women and women in love may suggest differential activation of the HPA axis in the early stages of romantic love, we found no differences in the cortisol levels of these two groups of women. Several studies have investigated the role of cortisol in the beginning of a romantic relationship and have reported mixed results. Some studies report an increase in serum basal cortisol levels during the beginning of a romantic relationship [1], consistent with research on non-human animals (e.g., prairie voles [32]), hypothesized to reflect the stressful conditions or arousal associated with the initiation of a new social contact, in turn promoting attachment [1,43,44]. Furthermore, Loving et al. [45] reported an increase in salivary cortisol levels in women when asked to think of their loved ones; however, the effect was mostly detectable in women characterized by a high amount of relationship-focused thinking. Maestripieri et al. [46] found that singles exhibited the highest cortisol response, individuals in short-term relationships showed mid-level cortisol levels relative to the group mean, and those in long-term positive relationships had the lowest cortisol levels, suggesting incremental changes in cortisol levels with relationship duration. In contrast to these studies, others have found lower daily salivary cortisol production in individuals during the beginning of a romantic relationship, associated with greater social reciprocity and partnership between new lovers and reported commitment to the relationship [4]. Indeed, Weisman et al. [4] showed that as early as two weeks after the initiation of a romantic bond, romantic attachment exerts positive effects on HPA functioning, lowering stress and cortisol levels. 

These inconsistent findings on the plausible effects of relationship initiation on HPA functioning (i.e., stress hormone levels) may be related to the methods of hormone measurement, e.g., plasma ([1]; and our study) versus saliva [4,45] or reactive cortisol [3,45] versus diurnal [4] or basal [1] levels. Furthermore, it may be an effect of comparing individuals in long-term relationships with singles (e.g., [1]) as controls for new lovers, which may in fact characterize opposite poles of the cortisol response, with new lovers in the middle [46]. Additionally, the level of passion and commitment toward a new partner is rarely controlled. Importantly, Weisman et al. [4] showed that commitment in a relationship, not passion, is the strongest predictor of changes in cortisol levels. It is also worth noting that an individuals’ hormone–behavior interaction might also be impacted by the partner’s hormone levels. For instance, some studies have documented that the relationship between cortisol and empathy, or testosterone and hostility, varies depending on the levels of these hormones in romantic partners [6]. Thus, future research may examine associations between an individual’s hormones and passionate love in the context of their partner’s hormonal status. 

As a limitation of the study, hormone levels were sampled from each woman on only one occasion (between the second and fourth day of the menstrual cycle), whereas an average from several time points may have been more representative. Moreover, although the sample size used in the present study was computed as sufficient to detect effects of an average size and was much larger than that used in previous research [1], it is possible that employing a larger and more representative sample would make our findings less speculative. As an additional limitation, we did not control for variation in sexual interactions of the two groups of women, which could potentially explain some of the variance in hormone levels between coupled and single women. 

## 5. Conclusions

To summarize, the results of this research suggest that romantic love may be related to hormonal changes in women, which may not only facilitate pair bonding and commitment, as shown in previous research [5,6,7,8,9,10], but could also potentially increase a woman’s fecundity. As such, physiological changes that occur when falling in love may be adaptive and may potentially increase a couple’s probability of conceiving offspring [47]. This adaptive interpretation may, at least partially, explain why romantic love is observed ubiquitously across diverse human cultures [48,49,50].

## Figures and Tables

**Table 1 ijerph-16-04224-t001:** Comparison of estradiol (E2, and E2/T), free testosterone (fT), prolactin (PRL), gonadotropin (FSH, LH), and cortisol (CT) levels between single women (*N* = 69) and women in love (*N* = 47). Descriptive statistics (mean, standard deviation, and range) and pairwise comparisons are also given for control variables (age and BMI). Means, SDs, and ranges are given for non-transformed values.

	Single Women (*N* = 69)			Women in Love (*N* = 47)			*t* (114)	*P*
	*M*	*SD*	Range	*M*	*SD*	Range		
E2 [pg/mL]	37.88	36.26	8.6–167.4	39.55	36.12	8.4–151.0	−0.41	0.68
fT [pg/mL]	1.21	0.73	0.3–3.5	0.92	0.59	0.3–3.1	2.52	0.01
E2/T	43.33	53.24	3.7–305.9	64.58	92.93	7.8–510.9	−1.90	0.06
PRL [ng/mL]	11.29	5.41	3.5–34.4	11.24	5.20	2.2–27.8	0.24	0.25
FSH [mi/mL]	6.71	2.44	1.2–15.4	7.55	2.35	3.7–13.7	−2.05	0.007
LH [mu/mL]	6.00	2.91	0.1–12.9	7.18	2.64	2.2–13.8	−2.39	<0.001
CT [ng/mL]	293.21	78.98	145.5–473.9	269.19	67.34	141.7–401.1	1.54	0.13
Age [years]	27.24	3.18	24–33	27.56	2.68	23–35	1.41	0.22
BMI [kg/m^2^]	22.68	3.18	15.8–32.0	21.95	2.68	18.1–28.7	1.29	0.20

**Table 2 ijerph-16-04224-t002:** Results of regression analyses examining the relationships between romantic relationship status (single vs in love) and hormone levels, controlling for age and BMI (*N* = 116).

Dependent Variable	Predictors	*Β*	*t* (112)	*P*
Model 1: *R*^2^ = 0.006, *F* (3,112) = 0.22, *p* = 0.88				
E2	Love status ^1^	0.04	0.43	0.66
	Age	0.03	0.36	0.72
	BMI	−0.05	−0.53	0.60
Model 2: *R*^2^ = 0.06, *F* (3,112) = 2.23, *p* = 0.09				
FSH	Love status ^1^	0.17	1.81	0.07
	Age	0.05	0.58	0.56
	BMI	−0.14	−1.51	0.13
Model 3: *R*^2^ = 0.07, *F* (3,112) = 2.72, *p* = 0.048				
LH	Love status ^1^	0.20	2.15	0.03
	Age	0.08	0.85	0.40
	BMI	−0.13	−1.37	0.17
Model 4: *R*^2^ = 0.02, *F* (3,112) = 0.88, *p* = 0.45				
PRL	Love status^1^	−0.04	−0.45	0.65
	Age	0.03	0.35	0.72
	BMI	−0.15	−1.60	0.11
Model 5: *R*^2^ = 0.08, *F* (3,112) = 0.33, *p* = 0.02				
E2/T	Love status ^1^	0.18	1.90	0.06
	Age	0.06	0.64	0.52
	BMI	0.06	0.62	0.54
Model 6: *R*^2^ = 0.02, *F* (3,112) = 0.88, *p* = 0.45				
fT	Love status ^1^	−0.24	−2.68	0.008
	Age	−0.05	−0.53	0.60
	BMI	−0.16	−1.75	0.08
Model 7: *R*^2^ = 0.02, *F* (3,112) = 0.88, *p* = 0.45				
CT	Love status ^1^	−0.15	−1.63	0.10
	Age	−0.10	−1.11	0.27
	BMI	−0.14	−1.49	0.14

^1^ Coded as single = 0, in love = 1.

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
