# Peer review of "Romantic Love and Reproductive Hormones in Women"

_ijerph, 2019, doi:10.3390/ijerph16214224_

Round 1
Reviewer 1 Report
This study compared serum hormone levels in the early follicular phase of the menstrual cycle in single women classified as not in love versus women reporting a high degree of love at the beginning of a romantic relationship. The results are provocative where serum LH and FSH are statistically significantly elevated (12.5% and 19.7%, respectively) while serum testosterone (T) is significantly decreased (24%) in women in love. No statistically significant differences were reported for serum estradiol (E2), cortisol, or prolactin between the two groups of women. Also of potential interest was a marginally significant (p<0.06) increase in the serum E2/T ratio (49%) for women in love. The study appears to be well designed and executed and the manuscript is very well written. The authors are commended for providing cautious interpretations of their observations, acknowledging conflicts with other reports, and providing reasonable explanations for these conflicts (e.g., regarding serum cortisol). This reviewer has no major concerns with the acceptability of this manuscript. In the opinion of this reviewer, the opening sentences of the Results section (lines 150-156) which describes the statistical methods employed should be in the Materials and Methods section. However, it is entirely up to the authors whether this suggestion should be incorporated. The observation of an increase in the serum E2/T ratio is provocative as it might be suggestive of increased aromatization of serum T by peripheral tissues. However, this possibility is highly speculative and the increase, though robust, fails to achieve statistical significance. Thus its incorporation at this stage is probably premature, but should be considered in future studies.
Reviewer 2 Report
The article submitted to the International Journal of Environmental Research and Public Health with the title “Romantic love and reproductive hormones in women” is within the scope of the journal. The topic of romantic love and the impact on endocrine regulation of the menstrual cycle in young-middle aged women is interesting and an uncommon subject of medical research.
Introduction: good language style, but especially first sentences are difficult to read and lengthy (e.g. line 40-45). The review of recently published articles to the topic is adequate.
Methods: The study design is simple and clear. Women were asked to participate to the study via advertisement in public media. The study inclusion criteria are clearly described and reasonable. The exclusion of women with diseases and acute infection is documented. The study sample of 116 women is compared to other studies on romantic love and endocrine regulation sufficient. Even though a calculation of study size is not presented - please add.
The inclusion criteria of romantic love was not only self-reported but also assessed using a subscale of a standardized and validated questionnaire on Love (STLS). It remains open if aspects of sexuality are also subject in the questionnaire. The endocrine blood tests were performed at the 2-4th day of the menstrual cycle to get standardized results of the endocrine menstrual regulation. Data on sexual interaction are probably not included.
Why was the “true study aim” concealed? How could the study design be explained and the informed consent taken by participants if the true study aim was concealed?
Please add more detailed comments on the statistical analysis (program, statistical testing, level of significance used). Clarify the measurement of testosterone as total or free testosterone, did you measure SHBG? Discuss the limited information of measurement of total testosterone.
The results show that testosterone seems to be significantly lower in women with romantic love than single women, FSH is significantly higher, LH is significantly higher. The regression analysis confirms the significant impact of love on FSH, LH, T and E2/T in this study design.
Discussion:
The discussion begins with a discussion of the lacking elevation of prolactin levels in women in the romantic love group. The possible relevance and effect of HPG axis activation on E2, T remains unclear. Why should an HPG axis activation resulting in LH/FSH increase be the basis of higher fecundity? The discussion continues with the possible effect on the regulation and function of the corpus luteum and thus luteal phase in women in love. Here further studies should be planned. Lower testosterone may result in increased fecundity but also in higher loyality and even monogamy of the female partner in the couple.
The information of a single cortisol testing in the serum is limited, this is discussed critically at the end of the article.
A summary at the end of the article is lacking. Very little is said about the topic of sexuality and romantic love in the partnership or sexual activity in the control group.
Reviewer 3 Report
The present study investigated in a representative sample of women, whether passionate love influences hormonal profiles in a way that can increase female fecundity. The authors compared blood serum levels of estradiol (E2, E2/T), luteinizing hormone (LH), follicle-stimulating hormone (FSH), prolactin (PRL), testosterone (T) and cortisol (CT) between single women (N=69) and women in love with their partner (N=47). The authors found that women in love had higher levels of gonadotropins (FSH, LH) and lower testosterone (T) levels compared to single women but did not differ in terms of estradiol (E2), prolactin (PRL) or cortisol (CT).
This is a very interesting research which grab the readers attention and get them interested in what the authors have to say. But there are some issues that need to be addressed before publication
In the methods section, the detailed introduction and description of statistics are needed, otherwise the results are not convincing. The statistical analysis of this study is relatively simple, and more advanced statistical analysis, such as correlation and Logistic regression analysis, will increase the expression of the results of this paper. More information should be added to Table 1, and I have marked in the attachment. For the regression analysis, in Table 2, why the author missed the variable "hsCRP" that presented in Table 1? Other suggestions have been marked in the attachment.

Round 2
Reviewer 2 Report
Thank you for the thorough revision of the paper, the revised paper has been improved substantially and should be published.
Reviewer 3 Report
I am very glad that the authors have carefully referred to my review comments, made due modifications, and supplemented the corresponding information. Statistical analyses have been greatly improved, although some cannot be done.The Tables have also been carefully modified. I think the revised manuscript has been greatly improved and is suitable for publication.